# Augmenting Ad-Hoc IR Dataset for Interactive Conversational Search

**Pierre Erbacher**                                                    *pierre.erbacher@isir.upmc.fr*

*Sorbonne Université*

**Jian-Yun Nie**                                                            *nie@iro.umontreal.ca*
*Université de Montréal*

**Philippe Preux**                                                       *philippe.preux@inria.fr*
*Inria, Université de Lille*

**Laure Soulier**                                                       *laure.soulier@isir.upmc.fr*
*Sorbonne Université*

**Reviewed on OpenReview:** *https://openreview.net/forum?id=XXXX*

## Abstract

A peculiarity of conversational search systems is that they involve mixed-initiatives such as system-generated query clarifying questions. Evaluating those systems at a large scale on the end task of IR is very challenging, requiring adequate datasets containing such interactions. However, current datasets only focus on either traditional ad-hoc IR tasks or query clarification tasks, the latter being usually seen as a reformulation task from the initial query. Only a few datasets are known to contain both document relevance judgments and the associated clarification interactions such as Qulac and ClariQ. Both are based on the TREC Web Track 2009-12 collection, but cover a very limited number of topics (237 topics), far from being enough for training and testing conversational IR models. To fill the gap, we propose a methodology to automatically build large-scale conversational IR datasets from ad-hoc IR datasets in order to facilitate explorations on conversational IR. Our methodology is based on two processes: 1) generating query clarification interactions through query clarification and answer generators, and 2) augmenting ad-hoc IR datasets with simulated interactions. In this paper, we focus on MsMarco and augment it with query clarification and answer simulations. We perform a thorough evaluation showing the quality and the relevance of the generated interactions for each initial query. This paper shows the feasibility and utility of augmenting ad-hoc IR datasets for conversational IR.

## 1 Introduction

Conversational systems, including personal assistant systems and chatbots, are becoming increasingly popular for a wide variety of tasks, including online information seeking. While more recent Large Language Models (LLM) like OpenAI's ChatGPT (Ouyang et al., 2022) have demonstrated their ability to answer factual questions, they cannot be considered as conversational search systems because they only generate the most likely answer text without referring explicitly to the sources and without guarantee of veracity, amplifying potential bias and false truth (called Stochastic Parrots) (Bender et al., 2021). To overcome this limitation, conversational search systems must rely on information retrieval capabilities to locate relevant sources/documents (Shah & Bender, 2022; Dalton et al., 2022; Zamani et al., 2022; Anand et al., 2020;

Bender et al., 2021). However, this does not mean that one can merely rely on models such as LaMDA (Thoppilan et al., 2022), WebGPT (Glaese et al., 2022) or Sparrow (Shuster et al., 2022) to generate an answer conditioned to the information retrieved by an independent tool, because top retrieved documents may not contain relevant information, leading to untruthful or uninformative answers (Nakano et al., 2021). This underlines the importance of including retrieval capabilities in the evaluation of conversational search models as a whole, as suggested in (Dalton et al., 2020a).

Beyond providing natural language responses, a key ability of conversational search systems is their (pro)active participation in the conversation with users to help clarify or refine their information need (Shah & Bender, 2022; Chu-Carroll & Brown, 1997; Dalton et al., 2022; Zamani et al., 2022; Anand et al., 2020; Bender et al., 2021; Radlinski & Craswell, 2017; Trippas et al., 2020; Aliannejadi et al., 2019; Keyvan & Huang, 2022; Zamani et al., 2020b). While recent advances in information retrieval based on LLM have significantly improved the performance of information retrieval (IR) models by reducing vocabulary mismatches between queries and documents (Formal et al., 2021; Reimers & Gurevych, 2019a; Hofstätter et al., 2021), accurately understanding the user's intent remains a challenge, in particular when the information need is complex, multi-faceted or when the resulting query is ambiguous (Culpepper et al., 2018). As the user cannot browse through the list of documents in conversational search, the conversational system must actively participate in the conversation and ask clarifying questions to help the user to clarify or refine their needs (Shah & Bender, 2022; Dalton et al., 2022; Zamani et al., 2022; Anand et al., 2020; Bender et al., 2021; Radlinski & Craswell, 2017; Trippas et al., 2020; Aliannejadi et al., 2019; Keyvan & Huang, 2022; Zamani et al., 2020b). This makes it particularly challenging to evaluate conversational search systems properly because the mixed initiatives of the user and the system can lead to many different directions.

Attempts have been made to evaluate conversational IR (Sekulić et al., 2021; Aliannejadi et al., 2019; Salle et al., 2021; Bi et al., 2021), but they mainly focus on evaluating the quality of the generation of clarifying questions using aligned datasets, such as Qulac (Aliannejadi et al., 2019) and ClariQ (Aliannejadi et al., 2021) that contain pairs of query and clarifying question. Other models (Hai et al., 2023a) address the retrieval task in the context of a conversation as in TREC CAsT (Dalton et al., 2020b). Owoicho et al proposed to use GPT4 to simulate user feedback on topics from the TREC CAsT datasets (Owoicho et al., 2023).

While the emerging trend highlights the need of designing ranking models taking into account mixed-initiative interactions, it also demonstrates the necessity to build large-scale IR datasets containing mixed-initiative interactions. The two datasets mentioned above, Qulac and ClariQ, have been built on the basis of queries of the TREC Web Track 2009-12 collection. However, they only contain at most 237 topics, limiting our ability to train and to test neural conversational retrieval models properly.

The critical need for large conversational IR datasets motivates us to carry out this study, which aims at creating large-scale mixed-initiative IR datasets, containing not only user's queries, but also user-system interactions. Collecting such conversational data is challenging because of not only the high annotation cost, but also the inherent difficulties to ask clarifying questions: the ambiguity arising from user's query depends on the context, the collection, and also the user's patience, expertise, cooperation, or engagement (Wang & Ai, 2021; Salle et al., 2021; Zamani et al., 2020a)

In our work, we aim to leverage simulation techniques to automatically generate mixed-initiative interactions between a user and a system and propose a methodology to augment ad-hoc IR datasets with such interactions. To do so, we design a query clarification generator leveraging the ClariQ dataset as well as a user simulation for user's response. We use them to generate mixed-initiative interactions on the MsMarco ad-hoc IR dataset. The resulting dataset is called MIMarco for Mixed Initiave MsMarco. Our contribution is threefold:

- We propose a methodology to augment ad-hoc IR datasets to integrate mixed-initiative interactions;

- We evaluate our dataset-building methodology, and particularly the quality of mixed-initiative interactions generated for the ad-hoc IR dataset MsMarco;

- We demonstrate the utility of simulated interactions for document ranking on augmented MsMarco. This result can also be seen as an evaluation proxy of the usefulness and the relevance of the simulated mixed-initiative interactions within the ad-hoc IR dataset.

## 2 Related Work

### 2.1 Evaluating Conversational Search

Designing an adapted framework for evaluating conversational search systems is still challenging in the IR community. Indeed, conversational search involves both dialogue and IR tasks that should lead to mixed-initiative interactions to support and guide the user during his/her search (Dalton et al., 2022). A conversational search system should be therefore able to 1) generate questions to clarify/elicit users' information needs, and 2) retrieve documents providing relevant information. There are two main strategies to train and evaluate these systems by either 1) leveraging existing datasets, often at the cost of not having all the dimensions of conversations, or 2) simulating interactions between the user and the system. We briefly review some typical attempts for different types of conversational systems.

On question answering (QA), the datasets have been extended from one-shot QA such as Squad (Rajpurkar et al., 2018), Quac (Choi et al., 2018), ELI5 (Fan et al., 2019), or OpenQA (Chen et al., 2017) to conversational Q&A such as coQA (Reddy et al., 2019). One can train/evaluate answer generation systems, and possibly information retrieval systems using the collection of passages. Despite this interesting move, the datasets are insufficient for IR because they usually focus on factual questions instead of complex or exploratory questions that characterize information needs. The TREC CAsT dataset (Dalton et al., 2020b) extends the scope of questions and addresses different information facets within the conversation (a facet can be seen as a specific sub-category of the topic). However, the interactions provided are often limited to answer generation without proactive interactions engaging the system in a real support of search. Other datasets, such as CANARD (Elgohary et al., 2019), focus on query refinement or reformulation, without proactive interactions. Therefore, most approaches focused on generating reformulated queries as input to ranking systems (Hai et al., 2023b). Last year, the TREC CAsT track (Dalton et al., 2020b) introduced mixed-initiative questions related to the proposed IR-oriented conversations, without providing the associated users' responses. This dataset constitutes a first step toward exploring mixed-initiative conversational search, but does not dispose of a complete and coherent conversation.

Several initiatives have been taken to build query clarification datasets (Rahmani et al., 2023). For example, the MIMICS dataset (Zamani et al., 2020a) contains large-scale open domain clarifying questions collected from real users on the Bing search engine. The query clarifications are associated with initial users' queries and other information such as clicks. However, this dataset does not provide document relevance judgments or conversational interactions between the user and the system. To the best of our knowledge, only Qulac and ClariQ datasets contain both document relevance judgments and the associated mixed-initiative conversations. These datasets are built from the TREC Web Track 2009-12 collection, which provides annotated topics and facet pairs, associated with relevant documents. Users' responses have been collected through crowd-sourcing platforms, allowing to build a complete dataset of mixed-initiative interactions grounded in an ad-hoc IR dataset. However, collecting these interactions has been costly and the datasets remain small with only 237 topics and 762 topic facets. This is too limited for training and evaluating conversational retrieval systems.

Facing the lack of adequate datasets, a growing idea in the community is to rely on user simulation to evaluate conversational search systems (Erbacher et al., 2022; Salle et al., 2021). User simulations by mimicking users' queries and feedback are cost-efficient and allow for the evaluation of various strategies without direct data annotations. For example, Salle et al (Salle et al., 2021) evaluate their query clarification systems with a user simulation aiming at generating answers. Their user simulation relies on a BERT model fine-tuned to classify "Yes"/"No" answers to the clarifying questions. With a controllable parameter, the user simulation can also add words from the intent in the answer to simulate more or less cooperative feedback. Sekulić et al (Sekulić et al., 2022) confirmed with an additional human judgment that user simulations can generate answers and give feedback with fluent and useful utterances. User simulation is also exploited to design

evaluation frameworks for conversational recommender systems (Kang et al., 2019; Gao et al., 2022; Wu et al., 2020; Zhou et al., 2020; Fu et al., 2020), resulting in large synthetic dialogue interactions from ad-hoc recommendation datasets (Kang et al., 2019; Gao et al., 2022; Wu et al., 2020; Zhou et al., 2020; Fu et al., 2020). However, in the recommendation context, we notice that conversations are generated under explicit search constraints over annotated features like price range, color, location, movie genre, or brand, whatever the generation approaches used – either by using large language models (Asri et al., 2016) or by following agenda (Schatzmann et al., 2007; Peng et al., 2018; Li et al., 2017; Kreyssig et al., 2018). Unfortunately, similar approaches cannot be used for complex and exploratory search tasks (Belkin & Croft, 1992). In open-domain IR, the facets underlying information needs are not necessarily discrete or easily identifiable, making it much harder to identify and annotate users' needs.

## 2.2 Asking Clarifying Question

Asking clarifying questions is a conversational task that allows the user to be involved in the query disambiguation process by interacting with the system. For open-domain conversational IR, a first line of works to identify a clarifying question relies on ranking strategies applied to a pool of predefined, human-generated candidates. In the pioneering work, Aliannejadi et al. (Aliannejadi et al., 2019) propose a ranker that iteratively selects a clarifying question at each conversation turn. Bi et al. (Bi et al., 2021) complete this approach with an intent detection based on negative feedback and a Maximal Marginal Relevance-based BERT. Hashemi et al. (Hashemi et al., 2020) use a transformer architecture to retrieve useful clarifying questions using the query and the retrieved documents. However, leveraging a fixed question pool will limit the coverage of topics, and therefore hinder the effectiveness of the approach. To overcome this limitation, a second line of works rather aims at generating clarifying questions. In (Salle et al., 2021), Salle et al. use templates and facets collected from the Autosuggest Bing API to generate clarifying questions. At each turn in the conversation, they select a new facet to generate the question until the user's answer is positive. This inferred facet is then used to expand the initial query. Sekulić et al (Sekulić et al., 2021) propose to further improve the fluency by using a LLM to condition the clarifying questions generation on the initial query and a facet. For instance, the query 'Tell me about kiwi', conditioned to facets 'information fruit' or 'biology birds' can generate questions like "Are you interested in kiwi fruit?' or 'Are you interested in the biology of kiwi birds?'. They rely on Clariq dataset to fine-tune GPT2, and found that generated questions are more natural and useful than template-based methods. They have extended this work by generating questions using facets extracted from retrieved documents (Sekulić et al., 2022). Zamani et al. (Zamani et al., 2020a) propose to generate clarifying questions associated with multiple facets (clarifying panels) which are collected using query reformulation data. They investigate three methods to generate clarifying questions: templates, weak supervision, and maximum likelihood, and use reinforcement learning to maximize the diverse set of clarifying panel candidates.
(Owoicho et al., 2023) The literature review highlights that there is a lack of adequate large-scale datasets containing mixed-initiative interactions for the IR task. Having in mind that collecting those datasets with human annotations would be costly, we believe that a possible alternative is to generate mixed-initiative interactions automatically from existing collections. We propose therefore a methodology to first generate clarifying questions and the associated answers, and second augment ad-hoc IR datasets. As already discussed earlier, the definition of facets in information retrieval is not always obvious but seems critical for generating relevant clarifying questions. We will put a particular attention on this aspect in our methodology.

# 3 Simulated interactions

## 3.1 Problem definition

We introduce our methodology to generate automatically large-scale mixed-initiative-driven IR datasets. To do so, we propose to augment ad-hoc IR datasets with simulated user-system interactions, namely clarifying questions (for the system side) and the corresponding answers (for the user side). To provide a dataset useful for training mixed-initiative-oriented neural ranking models and capturing similarity signals in the matching loss, it is important to provide a wide range of interactions, namely clarifying questions that give rise to either positive or negative answers. Having in mind that a topic might be complex or ambiguous,

we follow previous works (Sekulić et al., 2021; Zamani et al., 2020a; Salle et al., 2021) leveraging facets to generate those clarifying questions. Extracting positive or negative facets around a topic can be seen as a proxy to constrain the generation of clarifying questions expecting 'yes' and 'no' answers. Moreover, to ensure the overall quality of the mixed-initiative interactions, we propose to introduce another constraint variable modeling the user's search intent. The pair of facet and intent variables allows to generate positive and negative clarifying questions (thanks to the facet) by always keeping the answer generation coherent with the relevance judgments in the initial dataset (thanks to the intent). Said otherwise, sampling different facet-intent pairs from passages with known relevance judgment allows constituting a dataset with positive and negative mixed-initiative interactions that reflect the search intent of the user. For the sake of simplicity, we only consider single-turn interactions, and discuss the extension to multi-turn interactions in Section 6.

Let us consider an ad-hoc IR dataset $\mathcal{D} = \{\mathcal{P}, \mathcal{Q}, \mathcal{R}\}$, in which $\mathcal{P}$ is a collection of passages (or documents), $\mathcal{Q}$ is a set of queries, and $\mathcal{R}$ is a set of relevance judgments. $\mathcal{R}$ includes tuples $(q, \mathcal{P}_q^+, \mathcal{P}_q^-)$ indicating relevant $\mathcal{P}_q^+ \subset \mathcal{P}$ and irrelevant passages $\mathcal{P}_q^- \subset \mathcal{P}$, for a query $q \in \mathcal{Q}$. We assume $\mathcal{P}_q^- \cap \mathcal{P}_q^+ =$. Our objective is to augment this dataset $\mathcal{D}$ with a mixed-initiative interaction set $X = \{X_1, \ldots, X_i, \ldots, X_n\}$. We note a mixed-initiative interaction $X_i = (q, cq, a)$ where $q$ refers to an initial query, $cq$ a clarifying question, and $a$ the associated answer. With this in mind, we design a dataset-building methodology $\mathcal{M} : \mathcal{D} \to \mathcal{D} \cup \{X_1, \ldots, X_i, \ldots, X_n\}$ relying on two main steps: 1) extracting the (positive and negative) facets $f$ related to each topic (if not available in the initial ad-hoc IR dataset) which is then used to constrain the clarifying question generation, and 2) generating mixed-initiative interactions given a query $q$ and this facet $f$. Depending on the dataset, the sets of positive $\mathcal{F}^+$ and negative $\mathcal{F}^-$ facets associated with query $q$ might be available or need to be built (Section 3.2). We also assume that the search intent $int$ of the user is characterized by the relevant documents available in the initial dataset. Then we propose to generate a mixed-initiative interaction $X_i$ given a query $q$ and the constraint variables $f$ and $int$. We rely on 1) a clarifying model $\mathcal{CM} : q, f \to cq$ aiming at generating a clarifying question $cq$ around facet $f$ given the query topic $q$, and 2) a user simulation $\mathcal{US} : (cq, int, f) \to a$ which infer the answer $a$ in response to the clarifying question $cq$ given the relevance of the facet $f$ and the user intent $int$.

Below, we introduce the method to extract facets, as well as the components enabling to generate clarifying questions and answers. Then, we present the overall methodology to augment ad-hoc IR datasets with mixed-initiative interactions. Examples of expected mixed-initiative interactions are presented in Table 1.

## 3.2 Extracting Facet

Facets might be explicit or implicit depending on the dataset. For example, they are specified in TREC Web 2009-12 (Clarke et al., 2009), and accordingly, Qulac and ClariQ (Over, 2001)). If not explicitly specified, we propose to extract them from documents. Previous works have shown that query facets can be extracted from top-retrieved documents (Dou et al., 2016; Kong & Allan, 2013). Inspired by the analysis provided by Sekulić et al. (Sekulić et al., 2022), we extract top contextual keywords to represent facets, as suggested in (Sharma & Li, 2019). The goal of the facet extraction is to provide additional keywords that can be used to later generate a clarifying question about various topics or subtopics. In this work, facets are a set of keywords providing additional context to the query. We formulate it as a bijective function $\psi(P) :\to \mathcal{F}$ that maps a set $P$ of passages to a set of facets. Given a query $q$, we construct the sets $\mathcal{F}^+$ and $\mathcal{F}^-$ of positive and negative facets from respectively relevant and irrelevant passage sets, resp. $\mathcal{P}_q^+$ and $\mathcal{P}_q^-$. This allows us to keep the relevance of facets. To do so, for a passage $p \in (\mathcal{P}_q^+ \cup \mathcal{P}_q^-)$, we extract as a facet $f \in \mathcal{F}$ the set of K words in the passage that are the most similar to the passage embedding (i.e., the embedding of the [CLS] token). To compute the similarity, we use a pre-trained Sentence-Bert (i.e., MiniLM-L6-v2 model) (Reimers & Gurevych, 2019b) between each token embedding and the passage one.

## 3.3 Generating Mixed-Initiative Interactions

### 3.3.1 Generating clarifying questions

The goal of the clarifying model $\mathcal{CM}$ is to ask relevant clarifying questions relating to an ambiguity in the meaning or the object of the query. In most of the proposed models (Zamani et al., 2020a; Sekulić et al.,

| Simulated mixed-initiative interactions in MiMarco |
|---|
| **Query**: why did the us volunterilay enter ww1 
 **System**: do you want to know the effect of unrestricted submarine warfare on the us ? 
 **Answer**: yes 
 **Intent**: The effects of unrestricted submarine (U-Boat) warfare, economic ties to the Allies, the public outrage to the sinking of the Lusitania, the Sussex torpedoing, and the Zimmerman Telegram caused this change, and the United States entered an Allied Alliance and a World War for the first time in its history. 
 **Facet**: effects unrestricted submarine boat warfare |
| **Query**: find phone number for irs 
 **System**: are you looking for phone numbers for the irs internal revenue service? 
 **Answer**: yes 
 **Intent**: Internal Revenue Service Phone Numbers. 800-829-1040 For individual and joint filers who need procedural or tax law information and/ or help to file their 1040-type IRS Tax Help Line for individual returns (including Individuals\| Schedules C and E); and, general account information for Form 1040 Filers. 
 **Facet**: internal revenue service phone numbers |
| **Query**: lps laws definition 
 **System**: are you looking for a definition of a limited partnership? 
 **Answer**: no 
 **Intent**: The Court will not let you establish an LPS conservatorship unless it finds beyond a reasonable doubt, that the mentally ill person, is gravely disabled. Gravely disabled means that, because of a mental disorder, the person cannot take care of his/her basic, personal needs for food, clothing, or shelter. **Facet**: limited partnership business |

Table 1: Examples of simulated interactions belonging to the MiMarco dataset. In the first ad second example, both the intent and the facet are sampled from a relevant passage. In the third example, the intent is sampled from a relevant passage but the clarifying question is referring to a negative topic facet.

2022; Sekulić et al., 2021; Salle et al., 2021; Aliannejadi et al., 2019), this ambiguity is addressed by using the concept of facet. Therefore, the generation of clarifying questions $cq$ is conditioned on the initial query $q$ and a facet $f$:

$$p(cq|q, f) = \Pi_i p(cq_i|cq_{<i}, q, f) \tag{1}$$

where $q_i$ is the $i^{th}$ token in the sequence and $q_{<i}$ the previously decoded tokens. Our clarifying question generation is based on a pre-trained sequence-to-sequence model which is fine-tuned to generate a clarifying question $cq$ given the following input sequence:

$$\text{Query: q Facet: f} \tag{2}$$

where $Query$ : and $Facet$ : are special tokens.

In this paper, we limit the clarifying questions to those that expect yes/no answers.

### 3.3.2 User Simulation

The goal of the user simulation $\mathcal{US}$ is to mimic the user's answer in response to a clarifying question given his/her intent. In the user simulation, we expect accurate answers to clarifying questions, giving useful feedback to help the system understand his/her intent. The intent is a representation of the information need or of the goal behind the initial query. It is used to constrain the user simulation's answer towards this goal (Kang et al., 2019; Gao et al., 2022; Wu et al., 2020; Zhou et al., 2020; Fu et al., 2020; Erbacher et al., 2022). While sophisticated user simulations have been proposed to exhibit various types of behaviors like cooperativeness or patience (Salle et al., 2021), we limit the clarifying question to ask if the intent is about a facet and the answer of the user simulation to 'yes' or 'no' answer. This limited form of answer is motivated by two reasons: (1) despite the simplicity, a correct answer of this form corresponds to basic realistic interactions with users and is highly useful for the system to better identify the intent behind the

---

**Algorithm 1** Offline methodology for building Mixed-Initiative IR dataset

---

**Require:** $\mathcal{D} = \{\mathcal{P}, \mathcal{Q}, \mathcal{R}\}$
  $X \leftarrow \{\}$                                              $\triangleright$ Set of mixed-initiative IR-oriented interactions
  **for** $q \in \mathcal{Q}$ **do**
    $\mathcal{F}^+ \leftarrow \psi(\mathcal{P}_q^+)$                                        $\triangleright$ Extract the positive facets
    $\mathcal{F}^- \leftarrow \psi(\mathcal{P}_q^-)$                                        $\triangleright$ Extract the negative facets
    **for** $f \in (\mathcal{F}^+ \cup \mathcal{F}^-)$ **do**
      $cq \leftarrow \mathcal{CM}(q, f)$                              $\triangleright$ Generate the clarifying question
      **if** $f \in \mathcal{F}_q^+$ **then**                             $\triangleright$ Building the answer
        $a \leftarrow \text{'yes'}$
      **else**
        $a \leftarrow \text{'no'}$
      **end if**
      $X_i = (q, cq, a)$
      $X \leftarrow X \uplus X_i$                                   $\triangleright$ Increment the interaction set
    **end for**
  **end for**
  **return** $\mathcal{D} \cup X$

---

query. (2) This simple form of question and answer is easier to generate and evaluate. As an initial attempt, we prefer to start with this simple setting.

More formally, the user simulation aims at estimating the probability of an answer $a \in \{yes, no\}$ given a query $q$, a search intent $int$, and a clarifying question:

$$p(a|q, int, cq) \tag{3}$$

This is implemented as a sequence-to-sequence model that encodes the following input:

$$\text{Query: q Intent: int Question: cq} \tag{4}$$

and generates a 'yes'/'no' answer.

**Intent modeling** The user's intent corresponds to the user's information need and is only known by the user. While multiple intent representations can be adopted (such as a detailed description of the information need (Aliannejadi et al., 2019; 2021), a vector representation (Erbacher et al., 2022) or constraints (Kang et al., 2019; Gao et al., 2022; Wu et al., 2020; Zhou et al., 2020; Fu et al., 2020)), IR datasets usually do not have annotated intent associated with the query. However, relevant passages are known in an IR dataset. In this paper, we use a sampled relevant passage $p \in \mathcal{P}_q^+$ and assimilate its content to the underlying intent $int$. Formally: $int \leftarrow p$. We acknowledge that this choice relies on a strong hypothesis that passages annotated represent the user's search intent, we further discuss it in Section 7.

### 3.4 Leveraging Mixed-Initiative Interactions to Adapt Ad-Hoc IR Datasets

Given an ad-hoc IR dataset $\mathcal{D}$, our objective is to augment $\mathcal{D}$ with mixed-initiative conversations $X$. It is worth distinguishing the creation of training and testing datasets since they have different purposes. The training set requires including positive and negative interactions to allow the community to train properly mixed-initiative IR-oriented neural models. As a reminder, those positive/negative interactions are built on the basis of relevant and irrelevant documents determining positive and negative facets. Using the same heuristics to generate a testing dataset is not suitable since it would imply to include relevance judgments as evidence sources of the clarifying question generation at the inference step. Therefore, we propose to design an online evaluation methodology, leveraging the clarifying model $\mathcal{CM}$ and the user simulation $\mathcal{US}$ to generate mixed-initiative interactions without introducing a bias related to relevance judgments. We present these two methodologies aiming at generating offline and online datasets in what follows.

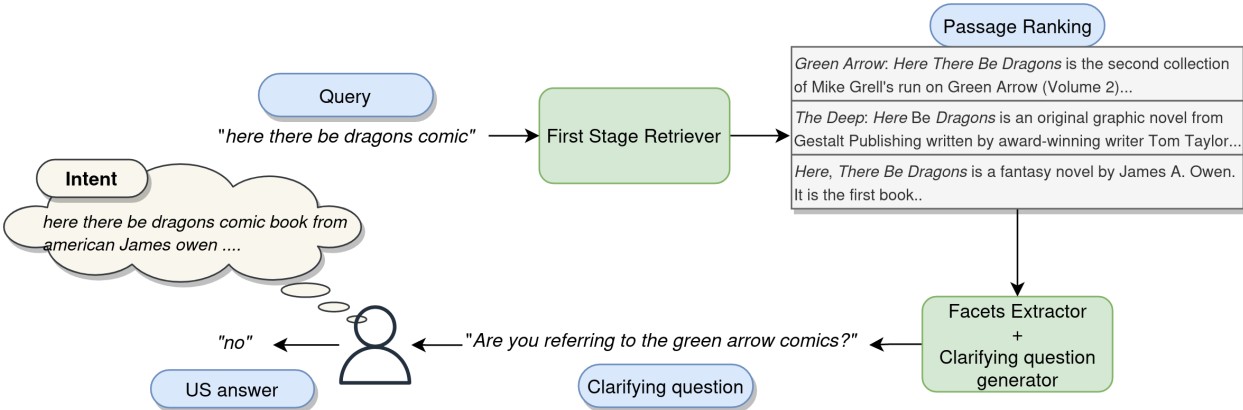

Figure 1: Online evaluation pipeline to create mixed-initiative interactions on a test ad-hoc IR set.

### 3.4.1 Building an offline training dataset with relevance judgments

Our offline methodology aims at generating a wide range of positive and negative mixed-initiative interactions on the basis of an ad-hoc IR dataset. To do so, we use relevant/irrelevant documents to build positive/negative facets constraining the clarifying question generation. As a supplementary quality constraint in the dataset supervision, we would like to ensure that answers fit with the relevance of the used documents. Said otherwise, the user simulation presented in Section 3.3.2 is replaced by a simple heuristic matching answers $a$ with the relevance of facets $f$:

$$a = \begin{cases} 'yes' \ if \ f \in \mathcal{F}^+ \\ 'no' \ otherwise \end{cases} \tag{5}$$

We propose the 3-step pipeline presented in Algorithm 1. Given a query $q$: 1) positive and negative facets, resp. $\mathcal{F}^+$ and $\mathcal{F}^-$, are extracted from relevant and non-relevant passage sets, resp. $\mathcal{P}_q^+$ and $\mathcal{P}_q^-$; 2) a mixed-initiative interaction $X_i$ is issued for a facet $f$, generating the associated clarifying question $cq$ (with $\mathcal{CM}$) and associating answer $a$ with the facet relevance (Equation 5); 3) the interaction set $X$ is incremented with this new interaction $X_i$, allowing to build a mixed-initiative IR dataset by associating the interaction set $X$ built over all queries with the initial ad-hoc IR dataset $\mathcal{D}$.

### 3.4.2 Building a testing dataset for online evaluation without relevance judgments

Our online methodology aims at generating mixed-initiative interactions without relying on relevant/irrelevant documents. Instead, we leverage pseudo-relevance feedback by using SERPs of a first-stage ranking model as a proxy to extract query facets. Each facet conditions the generation of the clarifying question and the answer. More particularly, the proposed pipeline to generate online mixed-initiative interactions for a query $q$ is presented in Figure 1. It is built on the following steps: 1) ranking documents using a first-stage ranker (in our case BM25), 2) extracting the set of facets on the basis of pseudo-relevant/pseudo-irrelevant documents, and 3) generating the mixed-interactive interaction.

Depending on the evaluation needs, different choices can be made regarding facet extraction. One can extract a single facet from the top-retrieved document to perform a single retrieval step for a query (the strategy used in our experiments). Other tasks or evaluation objectives would require generating multiple facets, and accordingly, multiple mixed-initiative interactions. This can be done by identifying top/flop documents obtained with the first-stage ranking as pseudo-relevant/irrelevant documents; each document conditioning the facet extraction as described in Section 3.2.

|                                        | Train set | Test set |
|----------------------------------------|-----------|----------|
| Number of documents                    | 8M        | 8M       |
| Number of query                        | 500K      | 6980     |
| Avg number of interactions per query   | 38.5      | -        |
| Avg length of clarifying questions     | 11.0      | -        |
| Percentage of positive answers         | 26.7%     | -        |
| Percentage of negative answers         | 73.3 %    | -        |

Table 2: Statistics of the generated mixed-initiative IR dataset MIMarco.

## 4 Assessing the Quality of the Dataset Generation Methodology

In this section, we evaluate our methodology, and particularly, the quality of simulated interactions. Please note that we focus on augmenting the MsMarco dataset but our methodology is generalizable to any ad-hoc IR datasets.

### 4.1 Evaluation protocol

#### 4.1.1 Datasets

We focus here on the MsMarco 2021 passages dataset (Nguyen et al., 2016) which is an open-domain IR dataset containing 8.8M passages and more than 500K Query-Passage relevance pairs with approximately 1.1 relevant passages per query on average. MsMarco is commonly used to train and evaluate first-stage retriever and cross-encoder architectures (Thakur et al., 2021). We leverage the MsMarco passage dataset with mined Hard Negatives released by sentence-transformers (Reimers & Gurevych, 2019b) [1] to build our tuples $(q, \mathcal{P}^+, \mathcal{P}^-)$. Hard Negatives are passages retrieved using a state-of-the-art retrieval method, which are more closely related to the query. They allow us to generate more relevant questions and answers.

To train the clarifying model $\mathcal{CM}$, we use the filtered version of the ClariQ dataset proposed in (Sekulić et al., 2021) that maps clarifying questions with facets. All clarifying questions in this dataset are built so as to expect 'yes'/'no' answers. This dataset provides 1756 supervised tuples of (query-facet-clarifying question) for 187 queries.

To train the user simulation $\mathcal{US}$, we do not use the answers included in the ClariQ dataset for supervision since answers are verbose (sentences with detailed information). Therefore, we leverage half of the train set of the MsMarco dataset ( 250000 queries) to extract positive and negative facets as detailed in Section 3.2 and generate clarifying questions using the $\mathcal{CM}$ model. The supervison label related to answers is inferred as proposed in the offline evaluation (see Equation 5).

To build a final dataset including training and testing sets, we respectively apply the offline evaluation methodology (Algorithm 1) on the other half of the training set (not used to train the user simulation) and the online evaluation methodology (Figure 1) on the test set of the MsMarco dataset. For the offline evaluation, because the original dataset includes sparse annotations, i.e. some passages are actually relevant but not annotated as such, it might be possible that relevant documents are considered as irrelevant ones. This trend is however exhibited in the MsMarco train set which only includes one relevant document by query. Therefore, to ensure labeling consistency, we follow (Qu et al., 2021) and denoise hard-negative in the training set using a well-trained cross-encoder model [2] that captures similarities between passages.

For the online evaluation, we choose to generate a single mixed-initiative interaction based on the top-retrieved document to fit with our extrinsic evaluation task based on IR. We will release, upon acceptance, the complete generated datasets as well as the clarifying model $\mathcal{CM}$ and the user simulation $\mathcal{US}$ to allow the generation of additional interactions. Statistics of the obtained mixed-initiative IR dataset, called MIMarco, are presented in Table 2. Table 1 depicts some examples of simulated conversations generated from MsMarco queries.

---

[1] https://huggingface.co/datasets/sentence-transformers/msmarco-hard-negatives
[2] https://huggingface.co/cross-encoder/ms-marco-MiniLM-L-6-v2

### 4.1.2 Baselines and metrics

**Evaluating clarifying questions with automatic metrics.** We follow (Sekulić et al., 2021) and compare our clarifying model, denoted $\mathcal{CM}$, with 1) a template-based approach (*Template*). The template follows a predefined sequence concatenating facets: 'Are you looking for + Facet'. 2) $\mathcal{CM}w/oFacet$: the version of our $\mathcal{CM}$ model only conditioned on the query. This is in fact based on a T5 model trained as a machine translation model, which generates a clarifying question from the query only.

We evaluate the ability of $\mathcal{CM}$ to generate clarifying questions using references provided in the ClariQ test set. We consider the METEOR metric (Banerjee & Lavie, 2005) and the average cosine similarity between sentence embeddings (COSIM). METEOR is commonly used to evaluate machine translation output considering unigram recall and precision. At the sentence level, this has a good correlation with human judgments (Banerjee & Lavie, 2005). To compute the similarity score, we encode the questions using a well-trained MiniLM-L6-v2 (Reimers & Gurevych, 2019b). We use t-test to assess the significance of metric differences (***: p-value<0.005).
To evaluate if the generated questions on MsMarco are similar to their relative passage, we also compute the mean cosine similarity between clarifying questions and their retrieved relevant and non-relevant passages. We encode the questions using MiniLM-L6-v2 (Reimers & Gurevych, 2019b).

**Human evaluations on clarifying questions.** To compare and better assess the quality of a generated clarifying question on MsMarco, we performed a human evaluation. Given the initial user query and the passage used to generate the question, we asked annotators to evaluate the quality of 200 sampled clarifying questions among the three models (*Template*, $\mathcal{CM}w/oFacets$, and our $\mathcal{CM}$ model). To do so, annotators are asked to select a preferred clarifying question from the three suggestions displayed in a shuffled order for the following criteria:

- Usefulness: Assess if a question can help to better understand or refine the query by providing additional information or suggestions.

- Naturalness: Assess the question fluency and readability.

- Relevance: Assess whether a question is specific or related to the information contained in a passage.

Each annotator evaluated 20 different cases and, for each metric to identify the best model output. We recruited 10 evaluators. Each instance is evaluated by 2 annotators and we obtain a Kappa metric equal to 0.324 showing a fair agreement between evaluators. We also distinguished results for both positive and negative user's answers by sampling relevant and irrelevant facets.

**Human evaluations on answers.** A strong hypothesis in our method is that clarifying questions generated with facets extracted from relevant passages lead to positive answers while using irrelevant passages to generate negative facets intrinsically leads to negative answers. To validate this strong hypothesis, we have shown human evaluators different instances including a query $q$, a clarifying question $cq$, and the relevant passage $p$ used to build the facet $f$. For each instance, we asked human evaluators to answer with 'yes' or 'no' to clarifying questions. This human evaluation involves 10 human annotators for a total of 200 questions, with balanced relevant and non-relevant facets used to generate the clarifying question. Each instance is annotated by 2 humans. We obtain a Kappa metric equal to 0.472 showing a moderate agreement between evaluators. To validate our hypothesis, we set human answers as the reference, and we compare them with our auto-labeling method (namely, the user simulation $\mathcal{US}$) to calculate the accuracy metric.

### 4.1.3 Implementation details

For both $\mathcal{CM}$ and $\mathcal{US}$, we used the pre-trained T5 checkpoint available on the Huggingface hub (Raffel et al., 2020; Wolf et al., 2019). To finetune these two models, we used teacher forcing (Williams & Zipser, 1989) and a cross-entropy loss. For optimization, we use AdaFactor (Shazeer & Stern, 2018), weight decay, and a

|  | METEOR | COSIM |
|---|---|---|
| *Template* | 0.338*** | 0.643*** |
| $\mathcal{CM}w/oFacet$ | 0.326*** | 0.608 *** |
| $\mathcal{CM}$ | **0.557** | **0.812** |

Table 3: Evaluation of different clarifying models on the test set of ClariQ dataset. Significance two-sided t-test: *** indicates statistically significant difference between baselines and our $\mathcal{CM}$ model (p<0.0015)

|  | q | p+ | p- |
|---|---|---|---|
| cq+ | 0.675 | **0.721** | 0.450 |
| cq- | 0.521 | 0.450 | **0.685** |

Table 4: Mean cosine similarity between generated clarifying questions and their related passages on the train set. The $cq+$, $cq-$ denote respectively clarifying questions generated using positive and negative facets.

learning rate of $5.10^{-5}$ with a batch size of 64. Keywords embeddings are computed using an off-the-shelf pre-trained MiniLM-L6-v2 model (Reimers & Gurevych, 2019b). The number of extracted words is fixed to $k = 5$ for the overall experiments. For inference, we use nucleus sampling (p=0.95) for the $\mathcal{CM}$ and $\mathcal{US}$ models.

## 4.2 Evaluation of the generated interactions

### 4.2.1 Automatic evaluation.

Table 3 reports the effectiveness of the clarifying model on the ClariQ test set. Results show that our model $\mathcal{CM}$ significantly outperforms all baselines. The lower results obtained by the $\mathcal{CM}w/oFacet$ baseline highlight that a simple machine translation model is less effective than templates using facet terms. Facets are useful to constrain the clarifying model, and seq-to-seq models based on large language models are more natural than template-based methods. Facets are extracted from a relevant or irrelevant passage and used to generate clarifying questions. Table 4 reports the cosine similarity between embeddings of questions and respective passages. We observe that the similarity between clarifying questions and their related passages (in bold) is higher than that between the clarifying questions and the queries. This shows that the generated questions are not generic to the query but oriented toward the provided passages.

### 4.2.2 Human Evaluation

We report human evaluation of clarifying questions in Table 5. The $\mathcal{CM}w/oFacet$ fine-tuned without facet generates more natural questions than other baselines (preferred for 46.3% of the sample). The $\mathcal{CM}$ model fine-tuned with facet generates more useful and relevant questions, this model is considered as the more relevant by evaluators in 59.9% of the test sample. This shows that the retrieved facet in the generation helps generate more useful and relevant questions.

In the human evaluation of answers, we obtain an accuracy of 0.685 between human answers and automatic labeling of clarifying questions. There are multiple causes explaining the difference between human answers and auto-labeling. 1) Facet may not always capture correctly the information provided in a passage, leading to poor clarifying questions. 2) The $\mathcal{CM}$ model does not always generate a question oriented toward the provided facet and produce a reformulation of the initial query, therefore asking a question not related to a facet.

Concerning the accuracy, we acknowledge that this may be low. We observed that this discrepancy may arise from questions labeled as non-natural. The main limitation comes from automatic facet extraction. Some keywords may not be representative of the document topics. Regarding the kappa, the disagreement may come from various reasons; naturalness may play an important role in judgment as observed in the table 6. Additionally, annotators must always choose a preferred question, even if none of the proposed questions are actually useful given the query or if clarifying questions are similar, thus adding noise to the agreement.

|  | Answer | Naturalness | Usefulness | Relevance |
|---|---|---|---|---|
| | positive | 0.044 | 0.086 | 0.120 |
| Template | negative | 0.073 | 0.095 | 0.146 |
| | total | 0.119 | 0.181 | 0.267 |
| | positive | **0.243** | 0.195 | 0.077 |
| $\mathcal{CM}w/oFacet$ | negative | **0.220** | 0.140 | 0.056 |
| | total | **0.463** | 0.336 | 0.133 |
| | positive | 0.206 | **0.213** | **0.297** |
| $\mathcal{CM}$ | negative | 0.211 | **0.268** | **0.301** |
| | total | 0.417 | **0.481** | **0.599** |

Table 5: Results of the human evaluation on Msmarco-passage. The $\mathcal{CM}$ without facet produces more natural questions, however not as relevant as $\mathcal{CM}$ with facet.

| **Examples for human evalutation** |
|---|
| **Query** webster family definition
**CQ1**: Are you looking for Noah Webster (1758-1843) lexicographer?
**CQ2**: Would you like to know more about Webster family definition?
**CQ3**: Are you referring to the lexicographer Noah Webster (1758-1843)?
**Passage** Noah Webster (1758-1843) was a lexicographer and a language reformer. He is often called the Father of American Scholarship and Education. In his lifetime, he was also a lawyer, schoolmaster, author, newspaper editor, and an outspoken politician. |
| **Query**: what is venous thromboembolism
**CQ1**: Would you like to know more about venous thromboembolism?
**CQ2**: Would you like to know more about venous thromboembolism?
**CQ3**: Are you looking for venous thromboembolism?
**Passage**: Venous thromboembolism (VTE) is the formation of blood clots in the vein. When a clot forms in a deep vein, usually in the leg, it is called a deep vein thrombosis or DVT. If that clot breaks loose and travels to the lungs, it is called a pulmonary embolism or PE. Together, DVT and PE are known as VTE - a dangerous and potentially deadly medical condition. |

Table 6: Table showing examples of clarifying questions shown to human evaluators. Evaluators must assess the most natural / relevant and useful question.

## 5 Evaluation on IR Task

In this section, we propose to assess indirectly the quality of the generated dataset through an IR task. Indeed, previous works (Qu et al., 2020; Zhou et al., 2020; Li et al., 2018; Fu et al., 2020; Jia et al., 2022) have already used extrinsic tasks to validate a dataset. Therefore, we introduce a neural ranking model which estimates passage relevance scores based on the query and a mixed-initiative interaction. Our objective is twofold: 1) Applying this model to our generated dataset provides some insights on whether the clarifying question and the associated answer actually give useful feedback to better understand the underlying information need. The evaluation is based on the following assumption: if a ranking model using the generated interactions outperforms the one without them, the interactions are deemed relevant and useful. 2) We provide a first baseline for mixed-initiative IR tasks.

### 5.1 Neural Ranking Model Leveraging Mixed-Initiative Interactions

We propose a simple model based on a cross-encoder architecture which has been shown to be effective for IR task, especially when using large language models (Pradeep et al., 2021). Previous cross-encoder aims at predicting the relevance of a passage $p$ given a query $q$ $P(relevant = 1|q, p)$. Our model estimates a score

for passages based on the query, a clarifying question, and a user answer $(q, cq, a)$, i.e.

$$p(relevant = 1|p, q, cq, a) \tag{6}$$

Following (Pradeep et al., 2021), the above score is transformed to the log-probability of predicting (decoding) the true/false tokens, i.e.

$$s_p = \log p(true|q, p, cq, a) \tag{7}$$

Following (Pradeep et al., 2021), we use the MonoT5 model and integrate mixed-initiative interactions to estimate document scores. The input sequence is a concatenation of query, document, question, and answer separated by special tokens:

$$\text{Query: q Document: d Question: cq Answer: a} \tag{8}$$

## 5.2 Training details

We used a pre-trained MonoT5 checkpoint available on the Huggingface hub (Raffel et al., 2020; Wolf et al., 2019). We fine-tune this model on our train set in 1 epoch, using our methodology with teacher forcing and a cross-entropy loss. We consider a maximum sequence length of 512 and a batch size of 128 sequences. In order to properly learn to contrast between relevant and non-relevant passages given a question, we use in-batch negative answers.

For optimization, we use AdaFactor (Shazeer & Stern, 2018), weight decay, and a learning rate of $10^{-4}$. The model fine-tuning takes approximately 4 hours on 4 RTX 3080 (24 Go).

At test time, we perform a first-stage retrieval on the initial query using the pyserini (Lin et al., 2021) implementation of BM25. We then apply our model as a second-stage ranker with additional information. We set the number of retrieved documents to 100.

## 5.3 Metrics and Baselines

We use classical metrics to evaluate the document ranking quality, namely the normalized discounted cumulative gain (NDCG) at rank 1, 3, and 10; and the Mean Reciprocal Rank (MRR) at rank 10.

To evaluate the potential of our mixed-initiative dataset, we compare the performance of our model, noted **BM25+CLART5**, against the following approaches:

- **BM25.** BM25 is a well-known sparse first-stage retriever commonly used as a baseline (Thakur et al., 2021).

- **BM25 + RM3.** RM3 is a pseudo-relevance feedback method for query expansion. The query is expanded using expansion terms extracted from the top 10 retrieved documents. RM3 is a competitive baseline and is still used for benchmarking IR models (Thakur et al., 2021; Adolphs et al., 2022).

- **BM25 + MonoT5.** MonoT5 is a second-stage ranker pre-trained on the original training set of MsMarco, i.e. only queries and relevance judgments. This model achieves state-of-the-art performance on the beir leaderboard (Thakur et al., 2021) and is a natural baseline as BM25+CLART5 uses the same second-stage pre-trained model before fine-tuning it on mixed-initiative interactions. We compare the MonoT5 performances with and without interaction. Interactions are added in the models context, following the same pattern as for CLART5 as shown in the equation 8.

## 5.4 Effectiveness of mixed-initiative-oriented neural ranking

We present the results of our mixed-initiative-driven neural ranking model obtained on the online evaluation pipeline presented in Section 3.4 applied on the MsMarco test set (Table 7).

Table 7 highlights the fact that additional information helps BM25+CLART5 improve significantly all retrieval metrics on the mixed-initiative-augmented MsMarco dataset. For example, BM25+CLART5 increases

|  | MRR@10 | NDCG@1 | NDCG@3 | NDCG@10 |
|---|---|---|---|---|
| BM25 | 0.1840*** | 0.105*** | 0.1690*** | 0.228*** |
| BM25 + RM3 | 0.1566*** | 0.0807*** | 0.1386*** | 0.2021*** |
| BM25 + MonoT5 (without interaction) | 0.3522*** | 0.2398*** | 0.3457*** | 0.4034*** |
| BM25 + MonoT5 (with interaction) | 0.3413*** | 0.2302*** | 0.3381*** | 0.3946*** |
| BM25 + CLART5 | **0.3863** | **0.2788** | **0.3817** | **0.4327** |

Table 7: IR effectiveness on the MiMarco test set. ***: two-sided t-test w.r.t. BM25+CLART5. with p-value<0.005

the MRR@10 score by 0.034 point compared to BM25+MonoT5. Further analysis of the results on MsMarco shows that for 33.0% of the queries, the relevant passage is not retrieved in the top-100 by BM25, leading MRR@100 to 0.0. For 25.6% of the queries Monot5 and ClarT5 obtain the same MRR@10. Out of all the queries, BM25+CLART5 achieves a superior MRR@10 for 30.3% of them, whereas it yields a lower MRR@10 for 11.1% of the queries. Overall these results show that the feedback provided by the user simulation to the clarifying question is relevant and useful. It helps increase the ranking of relevant passages. This result indirectly confirms that the simulated interactions indeed encode relevant information to the underlying search intents, which is what real users would provide in conversations. Therefore, the proposed simulations are reasonable.

# 6 Complementary Experiments

## 6.1 Extension to Multi-Turn Interactions

In the previous section, we simulated one interaction given a single query $X = (q, cq, a)$ for the online inference. However, multiple different facets can be extracted from retrieved passages. This means that sequences of interactions $X_0, ..., X_t$ can be inferred by sequentially selecting different facets. While a new pool of passages could be retrieved using the last interaction, we only consider here facets from passages retrieved with the initial query. Each $t^{th}$ turn exploits the $t^{th}$ document in the document list by the first-stage ranking to build a facet and generate a clarifying question. Multi-turn interactions are therefore generated in a non-arbitrary order.

**Impact on the design of the neural ranking model.** We propose to extend the model to multi-turn re-ranking using multiple clarification turns around the same query. We evaluate passages using multiple interactions around the same search intent. At each time step $t$ a new score $s_d^t$ is computed for the passages in the same ranking using a single interaction. This score is computed using Equation 9 which predicts cumulative relevance scores at all interactions, i.e. the sum of relevance scores till time $T$. This score is used as the ranking score of a document following a sequence of interactions $X_t = \{q, cq^1, a^1, ..., cq^t, a^t\}$. $cq^t$ and $a^t$ are the clarifying question and the answer generated at timestamp $t$.

$$s_d^T = \sum_{t=0}^{T} \log p(relevant = 1 | q, p, cq^t, a^t) \qquad (9)$$

where $s_d^T$ is the score of document $p$ at time $T$. As the ranking is updated between turns, we select facets from the top retrieved passage at each time step. We evaluate the retrieval performance at different lengths of interactions, from $T = 1$ to $T = 5$. We also report ranking entropy (Shannon, 1948) as a measure of the system's confidence by measuring how the scores are distributed in the ranking. This entropy is maximized when the score distribution is uniform over the ranking.

**Results** Table 8 shows the ranking performance while increasing the number of clarifying turns for the MsMarco test set. We observe that increasing the number of turns does improve MRR, NDCG scores and while minimizing the Entropy, suggesting that additional feedback helps to refine the user's information need further.

| | MRR@10 | NDCG@1 | NDCG@3 | NDCG@10 | Entropy |
|---|---|---|---|---|---|
| BM25 + CLART5 T=1 | 0.3863 | 0.2788 | 0.3817 | 0.4327 | 2.951 |
| BM25 + CLART5 T=2 | 0.44467 | 0.35186 | 0.43734 | 0.48038 | 2.303 |
| BM25 + CLART5 T=3 | 0.48176 | 0.39828 | 0.47483 | 0.51089 | 2.163 |
| BM25 + CLART5 T=4 | 0.50861 | 0.43266 | 0.50321 | 0.53183 | 2.06 |
| BM25 + CLART5 T=5 | **0.52949** | **0.45946** | **0.52311** | **0.54814** | **1.99** |

Table 8: Effectiveness results on $T$ interaction turns issued on the MsMarco test sets.

We depict qualitative examples of successful and unsuccessful clarification sequences in Table 9 as well as examples of failure cases in Table 10. We can see in the first example that additional interaction helps better refine passage scores leading to better MRR@100, while the entropy decreases. In the last interaction, the entropy is very low, meaning that the score distribution is dense on a few passages. On the other hand, the second example is a failure case where relevant passages are not even retrieved. The failure cases we observe, where the interaction turns deteriorate the ranking, shows that generated interactions are still not perfect. There is room for improvement in the future.

| Multi-turn Conversations | mrr@100 | Entropy |
|---|---|---|
| **User**: consolidate function | 0.125 | 4.388 |
| **System**: do you want to know what human resources functions the county provides? 
 **User**: no | 0.14286 | 3.91 |
| **System**: are you looking for media files referenced to itunes? 
 **User**: no | 0.16667 | 2.74 |
| **System**: do you want to know how to consolidate data source? 
 **User**: no | 0.2 | 2.61 |
| **System**: do you want to know how to use excel to consolidate data? 
 **User**: yes | 0.5 | 0.09 |
| **User**: where was the bauhaus built | 0.04167 | 3.854 |
| **System**: do you want to know where the bauhaus was founded? 
 **User**: no | 0.25 | 3.52 |
| **System**: do you want to know where the gropius sommerfeld house was built? 
 **User**: yes | 0.5 | 0.69 |
| **System**: do you want to know where the bauhaus art school was built? 
 **User**: no | 1.0 | 0.000218 |
| **User**: here there be dragons comic | 0.16667 | 4.27 |
| **System**: are you looking for a graphic novel called the deep ? 
 **User**: no | 0.2 | 3.12 |
| **System**: are you referring to the green arrow comics? 
 **User**: no | 0.25 | 2.93 |
| **System**: do you want to know what lenox globe dragons means is? 
 **User**: no | 0.33333 | 1.81 |
| **System**: are you looking for a james owen american comic book 
 **User**: yes | 1.0 | 1.15 |

Table 9: Examples of successful cases. In this example, asking clarifying questions helps to refine the user's intent and better identify relevant passages. These examples show typical ambiguity in the retrieved document. For example, the Bauhaus can refer to the rock band, the art school, the prize ..... *here there be dragons comic* can also refer to multiple comics with the same name.

| Failure cases | mrr@100 | Entropy |
|---|---|---|
| **User**: what are tor | 0.0 | 3.776 |
| **System**: do you want to know if tor is free? 
 **User**: no | 0.0 | 3.612 |
| **System**: do you want to know more about tors transoral robotic surgery? 
 **User**: no | 0.0 | 3.586 |
| **System**: do you want to know how to connect to a computer network 
 **User**: no | 0.0 | 3.567 |
| **System**: are you looking for a web browser for tor onion? 
 **User**: no | 0.0 | 3.541 |
| **System**: do you want to know about tor anonymity? 
 **User**: no | 0.0 | 3.329 |
| **User**: what is the fastest production sedan | 1.0 | 3.484 |
| **System**: do you want to know what the fastest production sedan is? 
 **User**: yes | 0.1 | 3.463 |
| **System**: do you want to know what the fastest production sedan is? 
 **User**: yes | 0.1 | 3.416 |
| **System**: do you want to know what the fastest production sedan is? 
 **User**: yes | 0.090 | 3.346 |
| **System**: do you want to know what the fastest production sedan is? 
 **User**: yes | 0.083 | 3.251 |

Table 10: Examples of failure cases. In the first example, none of the relevant passages are retrieved, asking clarifying questions and re-ranking passages do not improve scores. In the second example, the CM fails to capture passage's facet and generates the same questions. Additionally this tends to decrease the scores.

| | MRR@10 | NDCG@1 | NDCG@3 | NDCG@10 |
|---|---|---|---|---|
| BM25 | 0.2634 | 0.1648 | 0.2354 | 0.3055 |
| BM25 + RM3 | 0.2483 | 0.1457 | 0.2177 | 0.2941 |
| BM25 + MonoT5 | 0.4422*** | 0.3314*** | 0.4190*** | 0.4764*** |
| BM25 +CLART5 T=1 | 0.4749 | 0.3725 | 0.4471 | 0.5012 |
| BM25 +CLART5 T=2 | 0.54674 | 0.46002 | 0.51072 | 0.552 |
| BM25 +CLART5 T=3 | 0.58267 | 0.50956 | 0.54204 | 0.57787 |
| BM25 +CLART5 T=4 | 0.6047 | 0.53911 | 0.563 | 0.59291 |
| BM25 +CLART5 T=5 | **0.62115** | **0.56286** | **0.57666** | **0.60469** |

Table 11: IR effectiveness on the augmented version of the Natural Question test set (3452 queries). ***: two-sided t-test w.r.t. BM25+CLART5 T=1. with p-value<0.005

## 6.2 Transferability of the Methlogy

To test the potential to transfer the generators trained on a dataset to other datasets, we apply the same clarifying model $\mathcal{CM}$ and user simulation $\mathcal{US}$ trained on MsMarco (as described in figure 1) to generate simulated interactions on a new Natural Questions dataset (Kwiatkowski et al., 2019). Results are presented in Table 11. The higher results obtained by our method BM25+CLART5 w.r.t. other baselines suggest that the generated mixed-initiative interactions can benefit the neural ranking model. In other words, the generators trained on a dataset can be transferred to another dataset to create reasonable simulations. The experimental results on Natural Questions are consistent with those on MsMarco. This result is particularly interesting, showing that our methodology can be used in inference of out-of-domain datasets. This opens the potential perspective of constructing generic simulators of mixed-initiative interactions for any ad-hoc IR dataset.

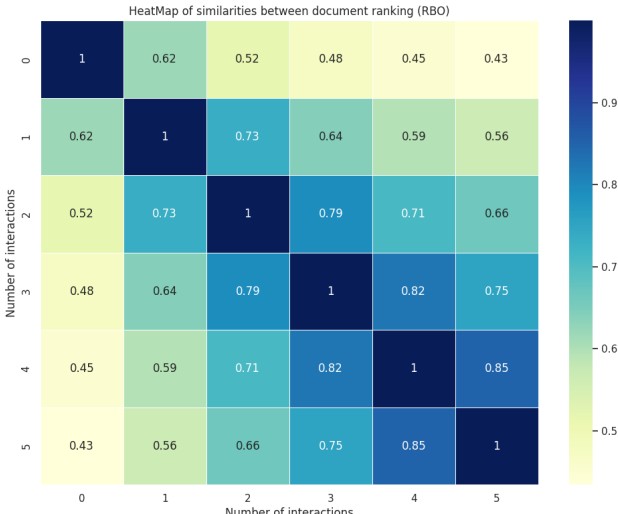

Figure 2: Passages ranking similarity between interaction turn. Ranked Biased Overlap (RBO) metric (p=0.9). Interaction 0 corresponds to the document ranking using Bm25 + MonoT5

### 6.3 Additional Analysis: Multi-turn Ranking similarity

In order to validate our results, we compute similarity metrics between ranking at each turn in the conversation. To measure the similarity between document rankings at different iterations, we rely on the Rank-Biased Overlap (RBO) (Webber et al., 2010):

$$\text{RBO}(S, T, p) = (1 - p) \sum_{d=1}^{\infty} p^{d-1}.A_d \tag{10}$$

where $S$ and $T$ are two document rankings, $d$ is the actual depth of the ranking. $A_d$ expresses the agreement (the size of the intersection of the two rankings) at depth $d$: $A_d = \frac{|S_{:d} \cap T_{:d}|}{d}$. $p$ determines the weight given to the top ranked document. We set $p = 0.9$, which means that the 10 top documents weigh 85% of the score.

RBO measures the similarity between incomplete and non-conjoint rankings and also values more heavily top ranked document. The more diverse the rankings, the lower the score. Figure 2 shows the ranking similarity for each additional user feedback. We can observe that the ranking seems to stabilize with the number of interactions: the similarity is higher between 4 and 5 interactions than between 0 and 1 interactions.

## 7 Conclusion and discussion

There is a critical need for adequate datasets with mixed-initiative interactions for conversational IR, but creating such a dataset is very costly. In this paper, we proposed a method to augment ad-hoc IR datasets by simulating a simple form of mixed-initiative interactions between a user and a conversational IR system. This method generates automatically clarifying questions and answers from a large open-domain IR dataset, making it possible to experiment conversational IR approaches at a large-scale. The proposed approach is generic and can be applied to any existing ad-hoc IR dataset. In the experiments, we augmented the MsMarco dataset and evaluate the quality of the interactions with intrinsic and extrinsic tasks, relying on automatic metrics and human evaluations. The results show that, despite the simple form, the generated interactions are relevant to the search intents and useful for better document ranking. This is a first investigation on large-scale dataset augmentation for conversational IR. It demonstrates the feasibility of the automatic construction of datasets. As a first investigation, this study has several limitations that can be improved in the future.

- First, our investigation is limited to clarifying questions based on a single facet, often assimilated to questions of the type: "Are you referring to 'facet'?". However, real clarifying questions might also question about multiple topics/facets in a single turn (ex: Are you interested to know about *topic1*, *topic2* or *topic3*) or also be formulated as open-ended questions (e.g., "What would you like to know about *topic*?"). These more complex questions are more difficult to generate and answer in simulations, but can potentially bring more information and be more natural in the conversation.

- Second, the facet extraction relied on a few keywords and this can be improved. We observe that when passages are long and address multiple topics, the generated question may not represent the topic addressed in the passage.

- Third, the user simulation has been limited to 'yes'/'no' answers. In a more sophisticated conversational search, the user might provide more and various information in the answer. Simulating more complex user's answers is a challenge for the future.

- Finally, we also generated multi-turn interactions but did not consider the dependency between turns. In real conversational search, later turns may depend on previous ones. More reasonable simulations of multi-turn interactions should take the dependency into account.

Despite the limitations, the demonstration of feasibility made in this paper to create large-scale conversational IR datasets opens the door for more investigations at large scale on the topic.

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

# A   Appendix

