# OpenReview forum: "Augmenting Ad-Hoc IR Dataset for Interactive Conversational Search"
_TMLR — Accepted by TMLR_

### Review · Reviewer_2xjA · 2024-02-12

**Summary Of Contributions:**

The authors propose a method to augment IR datasets with simulated interactions. These are based on having access to known "facets", or extracting them, and then asking the user if the query is about one facet or another. The simulated users only say "yes" or "no" for multiple interactions. It is shown that extracting additional info from the users in this way raises retrieval scores.

**Audience:**

Yes

**Claims And Evidence:**

Yes

**Requested Changes:**

These are references to weaknesses above.

Critical
---------
- Analysis of sequence length issues. If all of your samples are under 512, then no need.
- MonoT5 but with inputs to CLART5 as baseline for Tables 6 and 10.
- Careful editing of Page 14.

Nice to have
----------------
- Analysis of "denoising hard-negatives"
- Additional experiments that try other versions of conditionality on $p, q, cq, a$, if my comment is convincing to you
- Retraining with figure 1 instead of algorithm 1 to make training more in line with inference

**Strengths And Weaknesses:**

Thanks to the authors for the hard work on this paper.

Strengths
-------------
- Clear writing
- Good experimental results
- Big systems with a lot of thought-through designs

Weaknesses
----------------
- Sec 3.4.1 (algorithm 1), you use gold answers for $a$ when the question is about facet $f$. However, during inference time, you won't know the ground truth answer $a$. As such, you have a mismatch between training and inference. Why not use the pipeline in Figure 1 during training as well? Or a mix of Figure 1 and Algorithm 1. By the way, it's awkward to have these two presented in different formats. Figure 1 and Algorithm 1 should be presented alongside one another in a comparable format.
- Page 9: "denoise hard-negative in the training set" - can you please describe how this works and present some analysis? How important is this choice? Examples of what is left and what is removed?
- Sec 5.1 - formula (6), you always condition on $p, q, cq, a$. Since the user answer is just "yes" or "no", doesn't it make sense to either (a) exclude $cq$ and $a$ when $a$ = "no" OR (b) actually include the numerical value of $p(a | q, int, cq) in the posterior somehow? After all, if the probability of "yes" is 0.5, that is very different from it being "0.9".
- Sec 5.2 - re: maximum sequence length of 512. You are trying to cram a lot into a single instance with multiple back and forths etc. How often do you exceed 512? Is this an issue? Please include some analysis.
- Sec 5.2 - "we then apply our model as a second stage ranker" - to clarify, is the "model" what's in Figure 1? Be specific here.
- Page 14. Please reread this section and edit. Table 8 is missing a lot of values in the second column. Table 9 is not mentioned in the text.
- I feel like a natural baseline is missing. Consider table 6. CLART5 outperforms MonoT5, but it has more information, so this is not surprising, nor is it a "fair" comparison given the information disparity. What we don't know is whether MonoT5 can use this extra information without further training. As such, can you just feed the inputs of CLART5 into MonoT5 and see how well it does?  The same can be done for models in table 10.

---

> ### Author Response · Authors · 2024-03-19
>
> Thank you for your review, which provides great insights for us to improve on the quality of our paper.
>
> Regarding the mismatch between training and inference, the main objective of the paper was to propose a offline training/online inference because we don't have training data to simulate user simulation or train a ranking model. There are several reasons to train on clean and properly annotated documents. This allows us to have known "positive" and "negative" documents required for training the user simulation. Additionally, using documents labeled as relevant or irrelevant ensure less noise than using retrieval pipeline and pseudo relevance documents and using the online setup for training would induce additional computational cost because it would require retrieving and re-rank documents for each batch of example.
>
> Concerning the denoising of the training set, it is a classical methodology used in several IR papers [1,2]. Hard negative are irrelevant documents retrieved using a strong IR method (that have close representation as documents labelled as relevant). These  documents can be used in contrastive training as negative. However the training data of MSMarco is annotated sparsely meaning there is only one document annotated as relevant given a query while there may be other relevant documents. This is a known problem in the IR community, as it induces noise in the training (because false negatives may be used in the training as negative). To alleviate this problem a common practice is to denoise "hard negative" by removing false negative. A denoising method consists in relying on a strong cross-encoder (or re-ranker) to evaluate topK documents. Documents having a score higher than relevant documents are considered positive. This reduces noise and improves system accuracy.
>
> We acknowledge that hypothesis on user simulation may be simple, and we did not study the impact of probability of 'yes'. This may be very interesting to study how this impact feedback quality.
>
> Regarding the sequence size, there is no issue with the 512 sequence length as most MSMasrco passage are short and, clarifying question and query are short.
>
> The figure 1 shows the methodology to generate online user interaction. The experimental evaluation in Section 5.2 considers a downstream task leveraging the query q and the generated interaction (clarifying question cq, and the answer a) to re-rank documents using the model described in section 5.1.
>
> Concerning the remarks on Table 8 and 9, thank you for pointing out these issues, we will make corrections.
>
> Concerning the baselines, the goal of the ClarT5 is to measure if the synthetic data is actually useful.  The MonoT5 and CLART5 are very similar, they share the same architecture and the same training. The MonoT5 baseline is finetuned on MSMarco to perform classification on the query/document pairs. The CLART5 is a MonoT5 additionally finetuned on the MIMarco dataset with (query / clarifying question / answer / document) tuple, making it able to handle special tokens such as clarifying question and answer. We will add this baseline in the final version of the paper.
>
>
> [1] Qu et al 2020, RocketQA: An Optimized Training Approach to Dense Passage Retrieval for Open-Domain Question Answering
>
> [2] Formal et al 2022, SPLADE v2: Sparse Lexical and Expansion Model for Information Retrieval

---

> > ### Comment · Reviewer_2xjA · 2024-03-19
> >
> > Thank you for your response.

---

### Review · Reviewer_k81C · 2024-02-28

**Summary Of Contributions:**

In the context of IR dataset, the paper proposed a method to generate synthetic clarification questions and user interactions to make a dataset fit for interactive search training. The method involves the creation of a Clarification Model and a User Simulation. Given a query and a list of relevant + irrelevant documents retrieved, the method extract phrases as facets. Then a pretrained seq-to-seq deep model turn the facets into yes/no clarification questions (the Clarification Model) part. The User Simulation then give a binary answer to the clarification question based on the true intent; it's also implemented as a seq-to-seq model.

The paper mainly tests the methodology on the MsMarco dataset with two kinds of evaluations: directly on the quality of the generated clarification questions and simulated user responses, and indirectly by training a retrieval model on the augmented dataset and comparing it with baseline models. For direct evaluation, the paper showed similarity with reference golden questions from ClariQ, and also performed human blind evaluation. The result shows that model generated interactions are better than the baselines. For indirect evaluation, the paper finetuned a Mono-T5 pre-trained model to take the additional interactive data as input, and the model outperformed baselines, indicating that the augmentation of the dataset is effective in building better retrieval models.

The paper also performed additional evaluation by training a multi-turn retrieval model, which again showed stronger performance. The CM and US models are then moved to a new dataset, where they still show good performance, suggesting that the learning from MsMarco by the CM/US model can be reasonably transferred to a new dataset without retraining.

**Audience:**

Yes

**Broader Impact Concerns:**

No broader impact concern.

**Claims And Evidence:**

Yes

**Requested Changes:**

- There are different naming of the dataset, MsMarco and MiMarco. It's confusing and seems to be misspelling at the beginning, but later Reviewer realized that I in MiMarco might represent "Interactive". However a clarification of the name at the first mention (table 1)  is strongly recommended.
- Page 7 last sentence of Section 3.3: "... relies on a [strong hypothesis]  and we discuss it in Section 7". It's better to briefly summarize the hypothesis here to save the readers a trip to Section 7.
-  Section 5.4: "BM25+CLART5 obtains a better MRR@10 while [11.1%] obtain a lower MRR@10" is this a typo?
- Consider improving the baseline in Section 4.1.2. See the weakness section.
- Consider removing Section 6.3

**Strengths And Weaknesses:**

Strength
- Augmentation algorithm is completely human-free. This allows larger scale applications.
- The evaluation is very extensive. Reviewer appreciate the effort to have human blind test which adds more confidence than just similarity measurements.
- Additional evaluation on a different dataset to show the generalization capability of the method.

Weakness
- Baseline in 4.1.2 seems too weak. The CM w/o Facet baseline has no access to retrieved top documents, while still producing more fluence questions. A fair baseline would be to give the model at least some abstract of the retrieved documents.
- Section 6.3's goal is not clear. It shows the trend of increased similarity of ranking which is almost expected with the kind of iterative narrowing of search. The Reviewer see little additional value in that.
- Minor writing issues, could use more examples to help readers understand. See requested changes.

---

> ### Author Response · Authors · 2024-03-19
>
> Thank you for your review, which provides great insights for us to improve on the quality of our paper.
>
> Regarding the naming of the dataset, MsMarco is the original IR dataset and MiMarco is the augmented version to include mixed-initiative interactions (MiMarco for Mixed Initiative Marco). We thank you for pointing out this confusion, and we will update the name of the dataset.
>
> We note the several recommendations about the writing.
> - We will explicit the hypothesis in the section 3.3
> - We acknowledge that the sentence in section 5.4 may lack of naturalness. We will replace it by the following sentence: “Out of all the queries, BM25+CLART5 achieves a superior MRR@10 for 30.3% of them, whereas it yields a lower MRR@10 for 11.1% of the queries."
>
> With regard to the baselines, the aim is to measure how the facets improve the generation of clarifying questions. We followed the same experimental protocol as in the paper by Sekulic et al 2021, which provided supervision for clarifying questions. We acknowledge that the baseline may be weak, but template based methods and simple generative methods are still reasonable baselines due to the lack of available supervision. These baselines help us to assess and understand the performance differences between the aligned facets in the ClariQ dataset and the automatically extracted facets from the MSMarco dataset.
>
> Regarding the ranking similarity study in Section 6.3, the goal is to measure how much additional turns affect the ranking. We see value in these experiments because they confirm that document rankings are not similar to the initial document ranking.

---

> > ### Comment · Reviewer_k81C · 2024-03-31
> > **Thanks for the response.**
> >
> > I would note that my questions and requested changes are not all about the correctness of the paper, but also about how other readers could get the same confusion at the same places. So please make sure you incorporate necessary clarification / explanation at those places even if you decided not to change the paper (for example the baseline questions and Section 6.3's purpose).

---

### Review · Reviewer_RAMy · 2024-03-11

**Summary Of Contributions:**

The paper studies building an information retrieval system that can take initiative, specifically asking clarification question. They focus on building a large-scale datasets through simulation with LLM. Specifically, they fine-tune T5 checkpoint  to generate clarification question with ClariQ dataset.
They evaluate their approach by showing that simulated mixed-initiative interactions can help document ranking for MsMARCO, showing it outperforms baselines such as BM25 and its variants.
Overall this is an interesting direction, but I found the paper a bit hard to comprehend, and need iterations to be published. Please see detailed comments below.

**Audience:**

Yes

**Claims And Evidence:**

Yes

**Requested Changes:**

Here I provide some other comments:

Section 4.2: I find the accuracy of 0.685 to be fairly low, making me a bit concerned about the dataset quality. Concrete examples would be helpful.

Section 4.1.2: The Kappa of 0.324 is considered fairly low — what are some reasons for disagreements?



Table 2: what’s the unit of average length of clarifying questions?

Overall writings can be improved. Some concrete comments below:
* The last bullet point on introduction, “augmented” does not make sense to readers at this point.
* Please unify formatting, especially regarding single quote / double quote marks, these are used inconsistently throughout the paper.
* I think it would be helpful of define a few key terms - such as “facet” and “intent” a bit clearly earlier in the paper.
* At the end of Page 4 - “We assume P-q intersection P+q =? Is this missing something?

Algorithm 1:
This would be way more easier if you notate each component carefully in the figure caption.

Section 2.1: incorrect citation - QuAC is a multi turn QA setting, similar to CoQA.
Third paragraph — it would be helpful to clarify what’s the classification label space for these classification datasets.

**Strengths And Weaknesses:**

Strength:
* Motivation is strong.
* The evaluation framework is interesting and also covers extensions such as transfer to different dataset.

Weakness:
* The evaluation setting b/w single-turn vs. multi-turn interaction should be more clearly presented and discussed before we get to Section 6. Section 5.1, it assumes there’s only one clarifying question? This is a bit confusing as previously it assumed there could be multiple clarification questions per query? There could be multiple valid clarification question (one per facet?), how do you pick which facet to use for clarification question model?
* The data setup is unnatural: The dataset has an average number of 38.5 interactions per query — I think this setting is very unlikely? This is largely because of the design decision of having yes/no answer only clarification question..  This deserves a bit more discussion.
* Baselines are weak:
For evaluation on IR task: i think we should consider other baselines beyond BM25, such as dense retriever such as contriever or GTR?
The baselines for constructing clarification questions (template & CM w/o Facet) are both very weak — one is not even using full input. I am not sure this is very useful for readers. I am not sure this model is the main contribution of this paper -- very straightforward fine-tuning on supervised dataset w/ different prompt template than baselines?

---

> ### Author Response · Authors · 2024-03-19
>
> Thank you for your review, which gives us great insight into how we can improve the quality of our paper.
>
> Regarding the first weakness, there may be some confusion. Our goal is to generate training instances so that the clarifying question generator learns how to generate a clarifying question given a query. Therefore, we generate these instances using the facets extracted from the retrieved documents of each query in the dataset. This results in a maximum of 38.5 facets for a query, but it does not mean that we simulate 38 interactions in the conversation. In our information retrieval (IR) experiment, we consider only one interaction constructed from the top retrieved documents before re-ranking. In a second online scenario, we also consider up to 5 rounds of clarification between user and system, and we report ranking metrics as well as similarity between rankings. We will clarify this point.
>
> Regarding the baseline, in this work we compare both MonoT5 and ClarT5 with the same retriever in a re-ranking task, making BM25 an experimental choice rather than a baseline. The model generating the clarification questions is not a contribution of this paper, as described in the paper, we followed the existing method of generating clarification questions and applied the same evaluation protocol (Sekulić et al., 2021).
>
> Concerning the accuracy, we acknowledge that this may be low. We observed that this discrepancy may arise from questions labeled as non-natural. The main limitation stems from automatic facet extraction. Some keywords may not be representative of the document topics.
>
> Regarding the kappa, the disagreement may stem from various reasons; naturalness may play an important role in judgment. Additionally, annotators must always choose a preferred question, even if none of the proposed questions are actually useful given the query or if clarifying questions are similar, thus adding noise to the agreement.
>
> Example1:
>
> query:webster family definition
>
> Passage:Noah Webster (1758 â 1843), was a lexicographer and a language reformer. He is often called the Father of American Scholarship and Education. In his lifetime he was also a lawyer, schoolmaster, author, newspaper editor and an outspoken politician.
>
> cq1:are you looking for noah webster (1758 1843) lexicographer
>
> cq2:would you like to know more about webster family definition
>
> cq3:are you referring to the lexicographer noah webster (1758  1843)
>
> Example2:
>
> query: what is venous thromboembolism
>
> Passage:Venous thromboembolism (VTE) is the formation of blood clots in the vein. When a clot forms in a deep vein, usually in the leg, it is called a deep vein thrombosis or DVT. If that clot breaks loose and travels to the lungs, it is called a pulmonary embolism or PE. Together, DVT and PE are known as VTE - a dangerous and potentially deadly medical condition.
>
> cq1:would you like to know more about venous thromboembolism
>
> cq2:would you like to know more about venous thromboembolism
>
> cq3:are you looking for venous thromboembolism
>
> While in the first example the questions are dissimilar, in the second example the 3 questions are almost identical, making the preferred questions inconsistent between annotators.  We will add an example in the paper to clarify this point.

---

### Decision · Action_Editor_qJsi · 2024-04-26

**Recommendation:** Accept with minor revision

**Comment:**

There was no champion amongst the reviewers strongly in favor of either acceptance or rejection. The paper appears sound on a read-through, and the general sentiment across reviews was more weighed towards acceptance. Reading through the author replies, it seems to me that they have sufficiently addressed the main concerns of the sole dissenting voice amongst the reviewers, so that while there may be outstanding polemic issues which could be acknowledge in the camera ready, it is worth publishing. I recommend acceptance with minor revision. Should my recommendation be followed, I expect the authors to incorporate the feedback and the substance of their rebuttals and responses into their camera ready, and provide me with a list of changes made in order to obtain final approval.

**Audience:**

This should be of interest to a decently large crowd interested in producing conversational models and agents capable of asking clarificatory questions to resolve user information needs.

**Claims And Evidence:**

This paper proposes and evaluates a data augmentation method for producing training data which will reinforce the ability of models trained on it to pursue information retrieval-oriented conversations (i.e. conversational search) involving the asking of clarifying questions.